# Impulsivity, Emotion Regulation, Cognitive Distortions and Attentional Bias in a Spanish Sample of Gambling Disorder Patients: Comparison between Online and Land-Based Gambling

**DOI:** 10.3390/ijerph18094869

**Published:** 2021-05-03

**Authors:** Marta Sancho, Céline Bonnaire, Silvia Costa, Gemma Casalé-Salayet, Javier Vera-Igual, Rita Cristina Rodríguez, Santiago Duran-Sindreu, Joan Trujols

**Affiliations:** 1Addictive Behaviors Unit, Department of Psychiatry, Hospital de la Santa Creu i Sant Pau, 08025 Barcelona, Spain; scosta@santpau.cat (S.C.); gemmacs24@gmail.com (G.C.-S.); javierveraigual@gmail.com (J.V.-I.); sdurant@santpau.cat (S.D.-S.); jtrujols@santpau.cat (J.T.); 2Institut d’Investigació Biomèdica Sant Pau (IIB Sant Pau), 08025 Barcelona, Spain; 3Laboratoire de Psychopathologie et Processus de Santé, Université de Paris, F-92100 Boulogne Billancourt, France; celine.bonnaire@u-paris.fr; 4Youth Addiction Care, Center Pierre Nicole, Croix-Rouge Française, 75005 Paris, France; 5Department of Psychiatry, Bellvitge University Hospital-IDIBELL, 08907 L’Hospitalet de Llobregat, Spain; 6Mental Health Unit, Delta del Llobregat Primary Care Service, 08903 L’Hospitalet de Llobregat, Spain; cristinrcr@gmail.com; 7CIBER Salud Mental (CIBERsam), Instituto de Salud Carlos III, 28029 Madrid, Spain

**Keywords:** online gambling, gambling disorder, emotion regulation, attentional bias, impulsivity, gambling-related cognition

## Abstract

Several risk factors have been related to the onset and maintenance of gambling disorder (GD). The aim of this study was to explore the differences in emotion dysregulation, impulsivity, cognitive distortions, and attentional bias between online and land-based gamblers. The sample consisted of 88 treatment-seeking patients from the Behavioral Addictions Unit at the Hospital de la Santa Creu i Sant Pau, Barcelona (Spain). Patients were divided into two groups by considering their main type of gambling, i.e., online (*n* = 26) and land-based gambling (*n* = 62). Online gamblers were younger, more often employed, with a higher educational level than land-based gamblers. Regarding the rest of the variables, statistically significant differences were only found in *Positive Urgency*, one of the UPPS-P subscales, in which the land-based gambling group scored higher than the online gambling group. In this exploratory study, individuals with online and land-based GD phenotypes did not differ either in difficulties in emotion regulation or in attentional and cognitive biases. However, land-based GD patients showed a higher tendency to succumb to strong impulses under the influence of positive emotions. These preliminary findings warrant the need to continue investigating GD phenotypes in larger patient samples.

## 1. Introduction

In recent years, increased internet availability has brought changes in the way people gamble. Although land-based gambling is still the most frequent type of gambling, online gambling has grown considerably [1,2,3,4]. Previous studies have identified differences in sociodemographic characteristics between online and land-based gamblers. Online gamblers are more likely male, younger, non-Caucasian, single, well-educated, and employed than land-based ones [5,6,7,8]. Moreover, people who are involved in online gambling are more likely to be at risk of gambling, gamble more frequently, and present more severe gambling problems as well as a higher co-occurrence with other risky behaviors [9,10,11]. Nevertheless, these findings are not always consistent in the literature. Some authors [12,13] reported that land-based gamblers were more often employed than online gamblers, while Jimenez-Murcia et al. [13] found significant differences in education, socio-economic status, and the amount of money spent and debts accrued, but not in age, income, gender, employment, marital status, or gambling severity and psychopathology.

Different risk factors have been related to the onset and maintenance of gambling disorder (GD). Some of the most closely related factors are emotion dysregulation [14,15,16,17], impulsivity [18,19,20,21], attentional bias [22,23,24,25], and gambling-related cognitions [19,26,27].

### 1.1. Emotion Regulation

Regarding emotion regulation (ER) in GD, the literature has shown the importance of this construct in the development and course of GD [3,14,27,28]. Recently, Rogier and Velotti [16] proposed a gambling disorder conceptualization including the Gratz and Roemer’s ER model [29] and the ER process model developed by Gross and John [30] and extended by Sheppes et al. [31]. In this way, GD patients would show specific failures in the three main stages of ER processing (identification, selection, and implementation) accounted for the onset and the maintenance of the disorder. Other authors [28,32] proposed a GD classification based on the nature of ER motivations of the gambling activity. Nevertheless, ER motivations differ from ER processes (the former refer to the function of the disorder, the latter to the process implicated in the onset and maintenance of the disorder). Indeed, as suggested by Cole et al. [33], it is important to distinguish between the emotion regulation process and the emotions’ regulative function. To our knowledge, no study has investigated the ER process among online and land-based gamblers.

### 1.2. Impulsivity

Related to the above, both attention and impulse control are considered as part of the ER process [34]. In regard to impulse control, several studies have shown a relationship between GD and impulsivity, especially in its involvement in severity, poorer treatment outcome, and higher treatment dropout rate [35,36,37,38]. Recent work agrees on the importance of considering impulsivity as a concept with multiple components, which have different facets, underpinned by different psychological mechanisms [39]. According to the multi-dimensional of the impulsivity model UPPS-P (*Urgency, Premeditation (lack of), Perseverance (lack of), Sensation Seeking, Positive Urgency, Impulsive Behavior Scale*) [40,41], several studies found that patients with GD showed higher scores in the *Lack of premeditation* and *Urgency* dimensions, but not in the *Lack of perseverance* or *Sensation-Seeking* dimensions [18,42]. With regard to the *Urgency* dimension, patients with GD presented higher scores in *Positive* and *Negative Urgency* scales [26,32,43]. Furthermore, in relation to this emotional aspects of impulsivity, difficulties in regulating negative emotions seem to be the clearest marker of GD shared by all gamblers [16,26], whereas difficulties in regulating positive emotional states would be a characteristic of only some gamblers [16].

To our knowledge, few studies have compared online and land-based GD patients for the impulsivity dimensions. This literature reports contradictory results. Therefore, in some studies, online gamblers had similar impulsivity scores compared to offline gamblers [14], while in other studies, the online gambling subgroup showed a higher impulsivity than the land-based gambling subgroup [12,37], especially poker gamblers [44,45]. Theses discrepancies in the literature could be linked to the model of impulsivity used (e.g., Barratt’s model or Whiteside and Lynam’s model) but also to the absence of consideration of the type of gambling [46].

### 1.3. Gambling-Related Cognitions and Attentional Bias

Another risk factor associated with impulsivity and ER is gambling-related cognitions, which influence clinical status and gambling preference [26]. According to Michalczuk et al. [42], gamblers’ impulsive choice correlates with gambling distortions’ level. Furthermore, Ruiz de Lara et al. [27] found that cognitive biases were highly correlated with the emotional and motivational aspects of impulsivity. These characteristics have been associated with online gamblers, who also present stronger cognitions than land-based gamblers [37,47,48]. Therefore, those gamblers with high levels of impulsivity and gambling-related cognitions could display a greater severity of gambling behavior [19,32,49].

In addition to gambling-related cognitions, a large body of research has demonstrated the importance of attentional bias in addictive behaviors, including GD [23,50,51,52,53]. That is, a gambling-related stimulus can be processed automatically once it is detected, making it difficult to divert one’s attention from it. In line with previous studies, people with GD reported a faster detection of gambling-related stimuli [54,55,56,57]. Regarding differences in attentional bias among types of gambling, a limited number of studies have analyzed them. Gambling cues seem to increase subjective reports of craving in online gamblers [58], showing a strong attentional bias for gambling-related words [59].

### 1.4. Aims of the Current Study

Given the literature reviewed, the aim of this exploratory study was to compare online and land-based gamblers in a clinical sample regarding emotion regulation, impulsivity, attentional bias, and gambling-related cognitions. The choice to study treatment-seeking gamblers is based on data which shows that the severity of gambling behavior is higher in this population, besides having an impact on GD typology [60]. Moreover, the identification of possible GD clusters would be central to tailoring treatment for these patients. Taking this into account, we hypothesized that online gamblers would show higher difficulties in ER and higher levels of impulsivity, especially *Lack of perseverance, Negative Urgency* and *Sensation-Seeking*, as well as stronger attentional and cognitive biases.

## 2. Materials and Methods

### 2.1. Design and Participants

This exploratory study used cross-sectional data gathered from a treatment-seeking clinical sample before initiating psychotherapeutic treatment.

The whole sample consisted of 88 treatment-seeking patients diagnosed with GD according to DSM-5 criteria [61], who were, for the first time, attended to at the Behavioral Addictions Program of the Hospital de la Santa Creu i Sant Pau in Barcelona (Spain). The period of the recruitment was between September 2018 and September 2019. The current study was linked to the duration of a project funded by ONCE’s Fourth International Grant for Research on Responsible Gambling, Organización Nacional de Ciegos Españoles—ONCE.

Each participant was first assessed through a semi-structured face-to-face clinical interview and a subsequent psychometric evaluation, as part of the usual clinical practice and before being assigned to treatment. Psychologists with experience in behavioral addictions carried out the evaluation and diagnostic procedure, considering that it was part of the usual clinical practice. Patients were helped by clinicians in order to properly carry out the self-report measures and complete the computer test.

Inclusion criteria for the study were a minimum age of 18 years and meeting DSM-5 criteria for GD [61]. Exclusion criteria were the presence of a current psychotic disorder, a current affective episode, or an intellectual disability.

The study was approved by the Ethics Committee of the Hospital de la Santa Creu i Sant Pau (Barcelona) and was carried out in accordance with the Declaration of Helsinki principles. All patients accepted to be part of the study and signed an informed consent (IIBSP-PUB-2018-59).

### 2.2. Measures

#### 2.2.1. Socio-Demographic Characteristics

Socio-demographic variables (e.g., age, gender, marital status, educational level, employment status) were collected through a semi-structured face-to-face clinical interview.

#### 2.2.2. Gambling Behavior and GD Severity

Aspects regarding gambling behavior such as the main type of gambling and gambling mode were obtained from the information provided by each patient in the first clinical interview. Patients were diagnosed with GD, and their GD severity was estimated according to DSM-5 criteria [61].

#### 2.2.3. Impulsivity

The *Urgency, Premeditation (lack of), Perseverance (lack of), Sensation Seeking, Positive Urgency (UPPS-P), Impulsive Behavior Scale* [40,41] is a self-report questionnaire composed of 59 items which are scored on a 4-point scale (from agree strongly to disagree strongly). It assesses five different features of the impulsivity trait: *Sensation Seeking* (e.g., “I generally seek new and exciting experiences and sensations”), *Lack of Perseverance* (e.g., “I generally like to see things through to the end”), *Lack of Premeditation* (e.g., “I have a reserved and cautious attitude toward life”), *Positive Urgency* (e.g., “I tend to act without thinking when I am really excited”), and *Negative Urgency* (e.g., “I have trouble controlling my impulses”). A total score allows obtaining a global measure of impulsivity. The mean of the scores for all scales is 50 with a standard deviation of 10. The psychometrical properties of the Spanish adaptation are satisfactory [62], with a Cronbach’s alpha, in our study, ranging from 0.799 to 0.933 and a Cronbach’s alpha of 0.946 for all UPPS-P.

#### 2.2.4. Gambling-Related Cognitions

The *Gambling-Related Cognitions Scale* (GRCS; [63]) is a self-report questionnaire composed of 23 items and developed to assess five gambling-related cognitions, both in patients with a gambling disorder and in the general population. These five domains are the following: *Illusion of Control* (e.g “Praying helps me win”), *Interpretive Bias* (e.g., “Relating my winnings to my skill and ability makes me continue gambling”), *Predictive Control* (e.g., “I have some control over predicting my gambling wins”)*, Perceived Inability to Stop Gambling* (e.g., “I will never be able to stop gambling”), and *Gambling-Related Expectancies* (e.g., “Gambling makes me happier”). This instrument is composed of 23 items, with the total score ranging from 23 to 161. Higher scores suggest greater cognitive distortions. The Cronbach’s alpha for the *Gambling Expectancies* subscale was 0.687, showing for the rest of subscales a range of 0.703 to 0.789 and a Cronbach’s alpha of 0.883 for all GRCS.

#### 2.2.5. Emotion Dysregulation

The *Difficulties in Emotion Regulation Scale* (DERS; [29]) is a self-report questionnaire composed of 36 items which assess difficulties in regulating emotions. It is formed by six first-order scales: *Non-acceptance of emotional responses* (e.g., “When I’m upset, I become angry with myself for feeling that way”)*, Difficulties engaging in goal-directed behavior* (e.g., “When I’m upset, I have difficulty getting work done”)*, Impulse control difficulties* (e.g., “When I’m upset, I become out of control”)*, Lack of emotional awareness* (e.g., “I pay attention to how I feel”), *Limited access to effective ER strategies* (e.g., “When I’m upset, my emotions feel overwhelming“) and *Lack of emotional clarity* (e.g., “I am clear about my feelings”). A global measure of emotion dysregulation is obtained through a total score. Higher scores suggest greater problems with emotion regulation. The DERS psychometric properties are suitable, with a Cronbach’s alpha in our sample ranging from 0.709 to 0.912 and a Cronbach’s alpha of 0.936 for all DERS.

#### 2.2.6. Attentional Bias

The *Dot Probe Task* adapted for gambling is an attentional bias probe, which was adapted for gambling by our team, based on the original instrument [64,65,66]. It consists in the presentation of successive pairs of words on a computer screen, one related with gambling and the other neutral, after the appearance of a fixation cross for 500 milliseconds in the center of the computer screen. The position of the words was randomly chosen to be either above or below the location of the fixation cross. After a short time, the two words disappear, and a probe stimulus (X) appears in the location of one of the texts. Participants were asked to press one key if the probe was below and another if the probe was above the cross. This probe provides data about response latency, mistakes, and correct answers. The probe consisted of 80 trials (pairs of words), beginning with 10 practice trials. If the mean latencies of congruent trials were faster than those of incongruent ones, there was attentional bias. In this case, the result in the attentional bias variable, generated by the probe, was greater than zero. The probe administration was carried out with the Inquisit Lab 5.0 program and a MSI GP63 8RD-638ES laptop.

### 2.3. Statistical Analysis

The IBM SPSS 24 for Windows was used to perform the statistical analysis. A descriptive analysis of the main sociodemographic and clinical variables was performed. Measures of central tendency (mean) and dispersion (standard deviation) were used to describe continuous variables, while frequencies and percentages were used for categorical variables. The comparison between both groups (land-based vs. online) in sociodemographic variables was carried out with Chi-square tests for categorical variables and t-test procedures for quantitative variables.

One-way variance analysis (one-way ANOVA) was used to evaluate differences in clinical variables by groups (land-based vs. online). The statistical significance was defined as *p* < 0.05, unless otherwise stated. A Levene’s test was carried out to prove the equality of variances prior to the one-way ANOVA test. All one-way ANOVA tests met for equality of variances assumption. Effect sizes were reported as partial eta squared (*η*^2^_p_). Small, medium, and large effect sizes correspond to values of *η*^2^_p_ of 0.01, 0.06, and 0.14, respectively [67]. Taking into account the exploratory nature of the current study, as well as the previous literature related to it [68,69,70,71], corrections for multiple comparisons were not performed.

## 3. Results

### 3.1. Sociodemographic and Gambling Behavior Characteristics of the Sample

Patients were divided into groups, taking into account their main gambling mode: online (*n* = 26) and land-based gambling (*n* = 62). This classification was carried out according to the information provided by each participant both in the first interview and in the psychometric evaluation. Table 1 includes the description of the type of gambling for each group. Half of the patients with a preference for the online mode gambled on sports, followed by gambling on slot machines (30.77%), roulette (7.69%), stock exchange (7.69%) and cards (3.85%). On the other hand, 70.97% of the patients with a preference for the land-based mode gambled on slot machines, followed by roulette (16.13%), sports betting (6.46%), instant lottery (3.22%), bingo (1.61%), and cards (1.61%).

The online gambling subgroup (*n* = 26) was composed of 80.8% men and 19.2% women. More than half of the participants (57.75) were single, employed (80.7%), with a secondary (61.6%) or university (26.9%) educational level (see Table 2). The mean age of GD was 38.42 (SD = 11.251) (see Table 2), with a minimum of 21 years of age and a maximum of 64 years of age. The land-based gambling subgroup (*n* = 62) was composed of 85.5% men and 14.5% women, half of the participants were single (50%), employed (50%), with a primary (38.7%) or secondary (50%) educational level. The mean age of GD was 48.48 (SD = 15.377), with a minimum of 18 years of age and a maximum of 76 years of age.

Online gamblers were younger (*p* = 0.001), more often employed (*p* = 0.037), with a higher educational level (*p* = 0.023) than land-based gamblers (Table 2). No other significant differences were found between the groups for the remaining sociodemographic variables (*p* > 0.05 for all comparisons).

### 3.2. Comparison of GD Severity, Emotion Regulation, and Impulsivity Measures According to the Gambling Mode Preference

Table 3 contains the results of the ANOVA procedures for GD severity, emotion dysregulation, and impulsivity variables taking into account the gambling mode preference.

Considering the GD severity based on DSM-5 criteria [63], the results exhibited no significant differences between online and land-based gambling subgroups [F(1, 85) = 0.216; *p* = 0.643].

The results showed no significant differences between the two groups in the DERS subscales and in the total score (*p* > 0.05 in all subscales). Although online gamblers seemed to score higher in the *Impulse control difficulties* subscale (15.12 vs. 14.65) and *Lack of emotional awareness* subscale (17.60 vs. 17.23) than land-based gamblers and the latter apparently scored higher in the *Non-acceptance of emotional responses* (17.12 vs. 15.28), *Difficulties engaging in goal-directed behavior* (14.27 vs. 12.56), *Limited access to effective ER strategies* (19.33 vs. 16.84), and *Lack of emotional clarity* (13.70 vs. 10.20) subscales as well as in the overall score (96.47 vs. 87.60) compared to online gamblers, none of these seeming differences were actually statistically significant.

Regarding the UPPS-P subscales, the land-based gambling subgroup seemed to score higher in all subscales except for the *Sensation-seeking* subscale, in which the online gambling subgroup apparently showed a slightly higher score. However, these differences only reached statistical significance for the *Positive Urgency* subscale [F(1, 85) = 5.854; *p* = 0.018], with a moderate effect size (partial *η*^2^ = 0.064).

### 3.3. Comparison of Attentional Bias and Gambling-Related Cognitions Measures According to the Gambling Mode Preference

Table 3 includes the results of the ANOVA and Chi Squares procedures for attentional bias and gambling-related cognitions variables according to the type of gambling.

The results for the GRCS subscales and total score showed no significant differences between online and land-based gambling subgroups (*p* > 0.05 in all comparisons),. Although the online gambling subgroup seemed to score slightly higher in the *Gambling expectancies* (11.69 vs. 11.11) and *Interpretative bias* (11.31 vs. 11.23) subscales than the land-based gambling subgroup, while the latter scored apparently higher in *Illusion of control* (7.53 vs. 5.85), *Predictive control* (15.77 vs. 13.88), *Inability to stop gambling* (18.27 vs. 16.77), and GRCS total score (64.44 vs. 57.88), none of these marginal differences were actually statistically significant.

Regarding Dot Probe Task results, the land-based gambling subgroup seemed to show a higher proportion of correct answers than the online gambling subgroup (0.9603 vs. 0.9292) but also a greater presence of attentional bias, whereas the online gamblers showed apparently higher average latencies in both congruent and incongruent trials. However, as in most of the comparisons previously performed, these differences did not reach the level of statistical significance (*p* > 0.05 in all comparisons).

## 4. Discussion

The aim of this study was to explore differences and similarities between online and land-based GD gamblers seeking treatment, in socio-demographics, impulsivity, emotion regulation, attentional bias, and gambling-related cognitions. Of these 88 GD seeking-treatment patients, 26 showed a preference for online gambling, while 62 participants exhibited a preference for land-based gambling. Contrary to our hypotheses, only a few differences were statistically significant between the groups. The online gambling subgroup was younger, more often employed, with a higher educational level compared to the land-based gambling subgroup. Furthermore, land-based GD patients showed a higher *Positive urgency* than online GD patients.

In line with previous studies [3,12,72], the GD seeking-treatment patients with a preference for online gambling in our sample were significantly younger than the land-based ones. The widespread availability of online gambling, along with its strong advertising, may influence the greater involvement of the young population in online gambling [73]. This, together with risk factors inherent to youth, such as immaturity or poor impulse control, may contribute to the development of a higher severity of online gambling behavior [9]. In our study, online GD patients exhibited both an educational level and an employment rate significantly higher compared to land-based GD patients, without differences between the two groups in gender and marital status. A better educational level is consistent with previous findings (e.g., [5,74,75]), in contrast to the results found in our study regarding employment, gender, and marital status (e.g., [12,13]). This could be explained by differences in the samples included in the studies, which took into account both adolescent and adult populations, while our sample was only composed of adult participants. Furthermore, our samples were not large and were uneven in size, so these results have to be taken with caution.

Concerning ER, no statistically significant differences were found between the online and land-based GD groups. Difficulties in ER have been considered a transdiagnostic risk factor for several mental disorders, including GD [16,17]. In any case, few studies has examined the differences in ER between online and land-based gambling. For example, Goldstein et al. [76] found that online gamblers experienced more negative affect than land-based ones and gambled to cope with this emotional distress. Although the authors investigated gambling motives but not ER process, difficulties in ER have been positively related to gambling motives, especially *Non-acceptance, Lack of control, Lack of clarity*, and the total emotion dysregulation measure [77]. Other studies have also found that online gamblers become involved in gambling as a strategy to regulate their negative emotional states, such as boredom or fatigue, compared to land-based gamblers [76,78]. These findings suggest that the gambling behavior could be used by some gamblers as a coping strategy in the presence of negative emotional states, whereas other gamblers would use it as a regulatory strategy of positive emotions [16,79]. This could explain the results obtained in our study, namely, the presence of emotion dysregulation in both land-based and online GD patients, as well as the absence of differences between the two groups.

One of the facets of emotion dysregulation is difficulties in controlling impulses, highly related to GD [19,37,80]. Specifically, *Positive urgency, Negative urgency*, and *Lack of premeditation* are the dimensions of the UPPS-P model more strongly associated with GD [26,32]. Taking into account the gambling mode preference, some authors have reported that online GD patients seem to exhibit higher impulsivity compared to land-based GD patients [12,37]. Nevertheless, our study only found a significant difference between the two groups in *Positive urgency*, for which land-based gamblers scored higher compared to online gamblers. These results are in accordance with some studies that analyzed differences between different types of games. Slot machine gamblers seem to score higher in *Positive urgency* compared to other types of gamblers, including online gamblers [80]. The fact that 70% of land-based GD patients in our sample showed a preference for slot machines could explain the finding of a higher *Positive urgency* score for this group. Moreover, land-based gamblers may feel valued when they win, experiencing more positive emotions and favoring a greater impulsivity in the gambling behavior [81].

No statistically significant differences on gambling-related cognitions between the groups were found. Despite the absence of differences, both land-based and online GD patients showed strong cognitive biases, in line with results of previous studies [26]. As reposted in the literature, GD patients exhibit stronger cognitive biases compared to healthy controls [82,83,84,85]. Furthermore, some studies found an association between high emotion dysregulation, especially difficulties in controlling positive emotions, and strong gambling-related cognitions [27]. This could explain our findings, indicating that land-based GD patients showed a marginally higher but non-significant overall score on cognitive distortions than online GD patients.

In agreement with previous evidence, another important construct involved in the maintenance of GD is an attentional bias towards gambling-related stimuli [22,48,49,50,51]. In this sense, GD patients showed a tendency to identify faster gambling-related stimuli than neutral stimuli, compared to healthy control groups [22,54,55]. Moreover, as in cognitive distortions, the presence of attentional bias is frequently associated with impulsivity [27]. Although the literature on the matter is scarce, online gamblers reported a higher craving when they were exposed to gambling cues such as gambling-related words [54,55,58,59]. Contrary to previous research, our results show no differences in attentional bias between groups. Nonetheless, about half of both groups exhibited attentional bias towards gambling-related words, with a marginally higher—though non-significant—prevalence in land-based GD patients. In line with this, some research found that slot machine gamblers maintained their gaze longer on cues that matched their preferred form of gambling compared to poker gamblers [25].

Our results seem to indicate no differences between online and land-based gamblers, except for some socio-demographic variables and positive urgency. This could indicate the existence of transdiagnostic processes between different GD phenotypes, taking into account the relationship between the main measures included in the current study. If these results were replicated in future research, Third Wave Cognitive Behavioral Therapies could be the treatment of choice, especially for those GD patients with a higher level of emotion dysregulation, impulsivity, and strong cognitive and attentional biases [86,87,88].

### Limitations

The current study has some limitations that should be mentioned. The first of them is the small sample size and thus the low statistical power of the analyses. However, the results that reached statistical significance are rather robust. Secondly, it is also worth noting the unequal sizes of the two groups. Although the number of treatment-seeking online gamblers is growing, treatment seeking land-based problem gamblers continue to be the majority. In any case, the lack of balance between the number of participants in each group derives from and reflects a real-world study conducted in the context of routine clinical practice. Thirdly, both groups were mainly composed of males. It would be necessary to expand the knowledge regarding the GD characteristics of females. Thirdly, a mixed gambling subgroup was not included in our study. Future research should be aimed at exploring the differences between online, land-based, and mixed gamblers. Fourthly, the study was carried out in a single, public sector, tertiary hospital. Therefore, the generalizability of the findings to other types of treatment centers needs to be determined in future works. In any case, the present findings may likely be invariant in similar settings. Finally, corrections for multiple comparisons were not performed, which could be perceived as a limitation. However, because of both the exploratory nature of the present study and the fact that multiple testing adjustments control false positives at the potential expense of false negatives, we preferred to maintain the original level of statistical significance and regard our results as tentative [68,69,70,71].

## 5. Conclusions and Future Implications

Despite the aforementioned study limitations, mainly attributable to the exploratory nature and real-world conditions of the study, our results showed no major differences between online and land-based GD patients concerning emotion dysregulation, impulsivity, or cognitive and attentional biases. If these results were replicated in future research, they could indicate the existence of transdiagnostic processes between different GD phenotypes. In this sense, both a comprehensive assessment of the patients and an appropriate treatment are highly important. Furthermore, taking into account the relationship between the main measures included in our study, Third Wave Cognitive Behavioral Therapies could adequately address GD patients’ needs, especially those of patients with a higher level of emotion dysregulation, impulsivity, and strong cognitive and attentional biases [86,87,88].

## Figures and Tables

**Table 1 ijerph-18-04869-t001:** Description of the type of gambling for the online and land-based gambling subgroups.

Main Type of Gambling	Online (*n* = 26)		Land-Based (*n* = 62)	
	*n*	*%*	*n*	*%*
Slot machines	8	30.77	44	70.97
Bingo	0	0	1	1.61
Cards	1	3.85	1	1.61
Roulette	2	7.69	10	16.13
Sports betting	13	50	4	6.46
Stock exchange	2	7.69	0	0
Instant lottery	0	0	2	3.22

**Table 2 ijerph-18-04869-t002:** Comparison between online and land-based gambling subgroups regarding sociodemographic characteristics.

Sociodemographic Variables		Online (*n* = 26)		Land-Based (*n* = 62)		*p*
		*n/m*	%/SD	*n/m*	%/SD	
**Gender**	*Men*	21	80.8	53	85.5	0.581
	*Women*	5	19.2	9	14.5	
**Marital status**	*Single*	15	57.7	31	50	0.370
	*Married*	10	38.5	20	32.3	
	*Separated–divorced*	1	3.8	8	12.9	
	*Widow/Widower*	0	0	3	4.8	
**Educational level**	*Primary*	3	11.5	24	38.7	**0.023**
	*Secondary*	16	61.6	31	50	
	*University*	7	26.9	7	11.3	
**Employment**	*Unemployed*	1	3.8	7	11.3	**0.037**
	*Employed*	21	80.7	31	50	
	*Student*	1	3.8	2	3.2	
	*Inability to work*	2	7.7	9	14.5	
	*Retired*	1	3.8	13	21	
**Age (years-old)**		38.42	11.251	48.48	15.377	**0.001**

Note. m: mean. SD: standard deviation. Bold: significant comparison (0.05 level).

**Table 3 ijerph-18-04869-t003:** Comparison between online and land-based gambling subgroups regarding DERS, UPPS-P, GRCS, and DPT scores.

	Online (*n* = 26)		Land-Based (*n* = 62)		ANOVA			
	Mean	SD	Mean	SD	MD	*F*-stat	*p*	*η* ^2^ _p_
**GD Severity**	6.96	2.441	6.70	2.319	0.26	0.216	0.643	0.003
**DERS**								
DERS Non-acceptance emotions	15.28	6.43	17.12	7.24	1.84	10.209	0.275	0.014
DERS Goal-directed behaviors	12.56	4.13	14.27	5.03	1.71	20.241	0.138	0.026
DERS Impulse control difficulties	15.12	9.24	14.65	4.95	0.47	0.093	0.762	0.001
DERS Lack emotional awareness	17.60	9.23	17.23	5.03	0.37	0.056	0.814	0.001
DERS Emotion regulation strategies	16.84	6.48	19.33	8.21	2.49	10.826	0.180	0.022
DERS Emotional clarity	10.20	3.59	13.70	10.37	3.50	20.696	0.104	0.031
DERS Total score	87.60	24.39	96.47	27.14	8.87	10.994	0.162	0.023
**UPPS-P**								
UPPS-P Negative urgency	55.73	11.37	57.72	10.10	1.99	0.656	0.420	0.008
UPPS-P Lack of premeditation	51.50	13.45	54.82	10.10	3.32	10.602	0.209	0.018
UPPS-P Lack of perseverance	51.92	10.82	56.33	12.12	4.41	20.560	0.113	0.029
UPPS-P Sensation seeking	45.58	9.08	45.31	10.137	0.27	0.013	0.909	0.000
UPPS-P Positive urgency	52.65	10.19	59.13	11.90	6.48	50.854	**0.018 ***	**0.064 ^†^**
UPPS-P Total score	51.65	11.05	55.87	10.85	4.22	20.720	0.103	0.031
**GRCS**								
GRCS Gambling Expectancies	11.69	6.28	11.11	5.39	0.58	0.192	0.663	0.002
GRCS Interpretive Bias	11.31	6.424	11.23	5.104	0.08	0.004	0.950	0.000
GRCS Illusion of control	5.85	3.042	7.53	4.790	1.68	20.746	0.101	0.031
GRCS Predictive Control	13.88	8.590	15.77	7.927	1.89	0.991	0.322	0.011
GRCS Inability to stop gambling	16.77	8.330	18.27	7.324	1.5	0.713	0.401	0.008
GRCS Total score	57.88	26.279	64.44	20.314	6.56	10.593	0.210	0.018
**DPT**								
DPT Proportion correct answers	0.9292	0.200	0.9603	0.053	0.311	0.916	0.342	0.015
DPT Congruent average latency—Category 1	477.83	131.23	444.30	95.81	33.53	10.314	0.256	0.021
DPT Incongruent average latency—Category 1	464.34	148.49	445.09	101.70	19.25	0.370	0.545	0.006
DPT Congruent average latency—Category 2	463.25	146.86	449.86	102.53	13.39	0.180	0.673	0.003
DPT Incongruent average latency—Category 2	460.66	138.44	451.22	101.36	9.44	0.096	0.758	0.002
	*n*	*%*	*n*	*%*	*χ* ^2^	*df*	*p*	
AB_Category 1	9	42.9	23	53.5	0.638	1	0.424	
AB_Category 2	8	38.1	23	53.5	1.339	1	0.247	

Note. SD: standard deviation. MD: mean difference. df: degrees of freedom. DERS: Difficulties in Emotion Regulation Scale. UPPS-P: Urgency, Premeditation (lack of), Perseverance (lack of), Sensation Seeking, Positive Urgency, Impulsive Behavior Scaleç GRCS: Gambling-Related Cognitions Scale. DPT: Dot Probe Task. AB: Attentional Bias. * Bold: significant comparison (0.05 level). ^†^ Bold: effect size into the moderate range (*η*^2^_p_ > 0.06).

## Data Availability

The datasets generated for this study will not be made publicly available because the data used in this study are part of the hospital database and are subject to restriction to protect patients’ confidentiality.

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
