# Peer review of "Impulsivity, Emotion Regulation, Cognitive Distortions and Attentional Bias in a Spanish Sample of Gambling Disorder Patients: Comparison between Online and Land-Based Gambling"

_ijerph, 2021, doi:10.3390/ijerph18094869_

Round 1
Reviewer 1 Report
This is an interesting study on an important topic. The authors have done a good job in describing some key characteristics of on-line and off-line gamblers. However, I see some points that should be addressed before considering this paper for publication.
- The introduction section, although useful to the reader in that it explores the study variables, is in my opinion too lenghty. In some parts it is unfocused on the specific topic of the manuscript. I suggest shortening it if possible.
- The authors could consider adding a graphical representation of the study variables in the introduction section. This representation could help the reader to have a clearer idea of the factors associated with gambling.
- In the method section: the authors do not specify whether the subjects were following a treatment (pharmacological or psychological/psychiatric) before the recruitment. They only state that the recruitment took place before the assignment to treatment. That does not mean that the subject could have followed a previous intervention. This detail should be clarified.
- Was the clinical interview at paragraph 2.2.1 based on structured or semi-structured questions? Was it developed ad hoc for this study?
- When describing the measures I suggest giving item examples.
- The main problematic point in the paper is represented by the small and non-balanced sample size. The authors correctly acknowledge this issue in the limitation section, however, I suggest stressing this point also in the method section possibly citing references of previous studies that demonstrated the usefulness of small sample size studies. I wonder why the authors did not wait for the online sample to enlarge before looking into the data. Maybe the study was linked to a project and the project ended? If so (or if there are specific reasons) I suggest adding this information to the text.
- In the discussion section the authors refer to the incomplete cerebral maturation related to impulsivity. That is theoretically correct, however, the paper is not focused on adolescents. I suggest replacing reference 11 that is related to adolescents.
Reviewer 2 Report
Dear colleagues, I hope this message find you well.
Thank you for giving me the opportunity of reading the work “Impulsivity, emotion regulation, cognitive distortions and attentional bias in a Spanish sample of gambling disorder patients: Comparison between online and land-based gambling”, it has been a very big pleasure to collaborate reviewing this manuscript. The topic of this paper is very interesting and it seems necessary to delve it. However, there are several questions to improve before to publish it. I would suggest some changes:
- It is advisable to follow the Microsoft Word templateor LaTeX template to prepare the manuscript.
Introduction:
- The structure of the introduction is not clear. I recommended to divide it into subsections. For example, I recommend to create a specific subsection where describe aims specifically. Moreover, lines must be numbered.
- It should be justified why these variables are chosen (Impulsivity, emotion regulation, cognitive distortions and attentional bias) and not others. Is it an arbitrary decision of the authors? This could be done in the second paragraph of the introduction.
Method
- Socio-demograpchi*
- 2 Measures: I recommend adding sample items to each questionnaire
- p value must be in italics
Discussion
- It is not clear how online gamblers characteristics (younger, more often employed, with a higher educational level) could alter the findings. The sample is not large and uneven in size. Please describe if this can have significant effects.
- It is necessary to describe in more detail the practical and theoretical implications.
References
Please, you must follow the standards proposed by MDPI. For example:
- Name of the journals abbreviated and in italics.
